# Characterization of the *ddt1* Mutant in Rice and Its Impact on Plant Height Reduction and Water Use Efficiency

**DOI:** 10.3390/ijms25147629

**Published:** 2024-07-11

**Authors:** Banpu Ruan, Yaohuang Jiang, Yingying Ma, Menghao Zhou, Fei Chen, Yanli Zhang, Yanchun Yu, Limin Wu

**Affiliations:** College of Life and Environmental Sciences, Hangzhou Normal University, Hangzhou 311121, China; ruanbp123@163.com (B.R.); 2022011010014@stu.hznu.edu.cn (Y.J.); 2023111010053@stu.hznu.edu.cn (Y.M.); 2020111010022@stu.hznu.edu.cn (M.Z.); 20130019@hznu.edu.cn (F.C.); 20167009@hznu.edu.cn (Y.Z.); ycyu@hznu.edu.cn (Y.Y.)

**Keywords:** rice, drought tolerance, plant height, cytochrome P450, *DDT1*

## Abstract

Rice (*Oryza sativa* L.), a fundamental global staple, nourishes over half of the world’s population. The identification of the *ddt1* mutant in rice through EMS mutagenesis of the *indica* cultivar Shuhui527 revealed a dwarf phenotype, characterized by reduced plant height, smaller grain size, and decreased grain weight. Detailed phenotypic analysis and map-based cloning pinpointed the mutation to a single-base transversion in the *LOC_Os03g04680* gene, encoding a cytochrome P450 enzyme, which results in a premature termination of the protein. Functional complementation tests confirmed *LOC_Os03g04680* as the *DDT1* gene responsible for the observed phenotype. We further demonstrated that the *ddt1* mutation leads to significant alterations in gibberellic acid (GA) metabolism and signal transduction, evidenced by the differential expression of key GA-related genes such as *OsGA20OX2*, *OsGA20OX3*, and *SLR1*. The mutant also displayed enhanced drought tolerance, as indicated by higher survival rates, reduced water loss, and rapid stomatal closure under drought conditions. This increased drought resistance was linked to the mutant’s improved antioxidant capacity, with elevated activities of antioxidant enzymes and higher expression levels of related genes. Our findings suggest that *DDT1* plays a crucial role in regulating both plant height and drought stress responses. The potential for using gene editing of *DDT1* to mitigate the dwarf phenotype while retaining improved drought resistance offers promising avenues for rice improvement.

## 1. Introduction

Rice (*Oryza sativa* L.) stands as one of the world’s most vital staple crops, sustaining over half of the global population. Plant height is a critical agronomic trait that greatly influences yield and resistance to lodging in rice. Thus, elucidating the genetic and molecular underpinnings that regulate plant height is essential for breeding high-yielding varieties. The regulation of plant height in rice results from a complex interaction between genetic factors and environmental cues. Extensive genetic mapping studies have identified numerous quantitative trait loci (QTLs) and genes associated with plant height [1,2,3,4,5,6,7,8,9]. These loci frequently correspond to the genes involved in hormonal signaling, cellular elongation, and transcriptional regulation. Notably, the gibberellin (GA) pathway is a major regulator of rice plant height, with key genes such as *SD1* (Semi-Dwarf 1) playing critical roles in GA biosynthesis and signaling, thus influencing internode elongation and overall stature [10,11,12]. Additionally, genes in the brassinosteroid (BR) signaling pathway also significantly affect rice plant height [13,14,15,16,17,18,19,20].

Drought stress triggers a suite of molecular responses in rice, orchestrated by an intricate network involving genes, proteins, and signaling pathways. Central to the plant’s response to drought is the hormone abscisic acid (ABA), which regulates stomatal closure to reduce water loss through transpiration, thereby enhancing water use efficiency under drought conditions [21,22,23]. Beyond ABA signaling, rice utilizes a plethora of molecular mechanisms to withstand drought, including the synthesis and accumulation of osmoprotectants like proline, glycine betaine, and soluble sugars, which help maintain cellular osmotic balance and stabilize proteins and membranes during drought [24]. Additionally, rice activates antioxidant defense systems to neutralize reactive oxygen species (ROS) and protect cells from oxidative damage induced by drought stress, with enzymes such as superoxide dismutase (SOD), catalase (CAT), and peroxidases (POX) playing key roles in detoxifying ROS and maintaining cellular redox homeostasis [25]. In plants, cytochrome P450 enzymes play crucial roles in diverse physiological processes, including the biosynthesis of phytohormones, secondary metabolites, and defense compounds. These enzymes are instrumental in generating a broad spectrum of specialized metabolites, such as flavonoids, alkaloids, terpenoids, and phenylpropanoids, which are essential for plant growth, development, and adaptation to environmental challenges [26,27,28,29]. Notably, members of the CYP85 and CYP90 families catalyze critical steps in the biosynthesis of BR, a class of steroid hormones vital for various aspects of plant growth and development, including cell elongation, vascular differentiation, and stress responses [30,31]. Cytochrome P450 enzymes also contribute to the biosynthesis of phytoalexins and defense compounds, enhancing plant defense against pathogens and environmental stresses. For example, the CYP71 and CYP81 families are involved in synthesizing various alkaloids and glucosinolates, which serve as chemical deterrents against herbivores and pathogens [32,33]. Furthermore, these enzymes are integral to the production of essential plant pigments such as anthocyanins and carotenoids, which contribute to the coloration of flowers and fruits [34,35]. 

The integration of drought tolerance with semi-dwarfism in rice holds significant agricultural importance. Semi-dwarf varieties, exemplified by the well-known *SD1* gene, have revolutionized rice cultivation by reducing lodging and increasing yield potential. However, their reduced stature can sometimes compromise their ability to withstand environmental stresses, such as drought. By combining drought tolerance traits with semi-dwarfism, breeders can develop varieties that not only maintain high yield potential and lodging resistance but also exhibit enhanced resilience to water scarcity. This dual trait integration addresses the evolving challenges posed by climate change and water scarcity, ensuring sustainable rice production in diverse agroecological settings.

In this study, we delineate a rice *dwarf and drought tolerance 1 (ddt1*) mutant and elucidate the *DDT1* gene, corresponding to the previously reported P450 gene *CYP96B4*/*SD37*/*DSS1* [29,36,37]. By employing genetic localization and molecular markers, we aimed to confirm the specific role of the *DDT1* gene in regulating rice growth and response to drought stress. The *ddt1* mutant identified in this study exemplifies this dual benefit, offering insights into enhancing both drought resilience and agronomic performance in rice breeding programs.

## 2. Results

### 2.1. Phenotypic Analysis of ddt1

The *ddt1* mutant was identified through a detailed screening process following the application of EMS (ethyl methane sulfonate) mutagenesis to the *indica* rice cultivar Shuhui527. Relative to wild-type (WT) plants, the mutant displayed a pronounced dwarf phenotype, evident from the seedling stage through to maturity (Figure 1A,B). This reduction in height resulted from a shortened length of all internodes in the *ddt1* mutant (Figure 1C,D). Additionally, the *ddt1* mutant exhibited smaller grain size as well as reduced grain weight and grain weight per plant compared to WT (Figure 1E and Appendix A).

### 2.2. Map-Based Cloning of DDT1

To conduct a genetic analysis of the *ddt1* mutant, we crossed the mutant with the wild-type cultivar Shuhui 527. A phenotypic examination of the F_1_ progeny revealed characteristics similar to the wild type, indicating the recessive nature of the mutation. In the F_2_ population, consisting of 325 plants, 250 of them showed growth patterns similar to the wild type, while the remainder exhibited the mutant phenotype. The observed segregation ratio of 3:1 (χ²_0.05_ = 0.24 < 3.841) supports the hypothesis that the *ddt1* mutation is controlled by a single recessive allele at a nuclear locus.

To identify the gene associated with the mutant phenotype, the *ddt1* mutant was crossed with the *japonica* rice variety Wuyunjing 7. From this cross, 210 F_2_ progenies manifesting the mutant phenotype were selected for further analysis. We utilized 183 simple sequence repeats (SSRs) and 41 sequence tagged sites (STSs), uniformly distributed across the 12 chromosomes of rice, for genotyping. Initial mapping efforts localized the mutant locus between markers B3-1 and B3-3 on chromosome 3, as determined by analyzing 22 F_2_ plants (Figure 2A). Further fine mapping with a larger set of 210 F_2_ mutants refined the locus to a specific interval flanked by markers P3 and P4 (Figure 2A). According to data from the Rice Genome Annotation Project (RGAP), this interval contains six predicted open reading frames (ORFs). Sequencing of this region in the *ddt1* mutant revealed a transversion mutation from A to T within an exon of *LOC_Os03g04680*, encoding a cytochrome P450 enzyme, which resulted in a premature termination of the protein (Figure 2A). Therefore, *LOC_Os03g04680* was identified as the candidate gene responsible for the observed mutant phenotype.

To definitively confirm that *DDT1* corresponds to *LOC_Os03g04680*, we carefully inserted a genomic fragment of *LOC_Os03g04680* into the binary vector pCAMBIA1300 and introduced it into the *ddt1* mutant through a transformation. This yielded 12 independent T_0_ transformants. Detailed phenotypic analysis (COM-1, COM-2) demonstrated that each transformant effectively mitigated the *ddt1* mutant phenotype, which was evidenced by enhanced plant height (Figure 2B,C). These findings provide compelling evidence supporting the hypothesis that the dwarf phenotype observed in the *ddt1* mutant is due to a single-base substitution mutation within the *LOC_Os03g04680* gene.

Through qRT-PCR analysis, our study revealed a broad expression spectrum of the *DDT1* gene across various plant tissues, including the roots, culms, leaves, leaf sheaths, and panicles. Notably, *DDT1* showed its highest expression level in leaf tissue, while its lowest expression was observed in the panicle (Figure 2C).

### 2.3. Subcellular Localization of DDT1

To elucidate the subcellular localization of DDT1, our study utilized a strategic approach by creating a recombinant expression vector (p35S::DDT1-GFP). This vector was specifically designed to fuse the DDT1 coding sequence with the green fluorescent protein (GFP) marker, enabling visual tracking of the protein within cells. The construct was then introduced into rice protoplasts, which served as an effective model system for plant cell biology studies. Employing advanced confocal microscopy techniques, we carefully observed the distribution of the DDT1-GFP fusion protein within these cells. Our findings clearly demonstrate that DDT1 predominantly localizes to the endoplasmic reticulum (Figure 3), providing significant insights into its functional role in cellular processes.

### 2.4. Unveiling the Role of DDT1 in Plant Height Regulation

We investigated the expression patterns of genes involved in gibberellic acid (GA) synthesis and degradation in the wild type (WT), *ddt1* mutant, and complementary lines. Our findings demonstrated significant alterations in the expression levels of key genes: *OsGA20OX3* was notably increased, while *OsGA20OX2* was significantly decreased in the *ddt1* mutant compared to WT and complementary lines (Figure 4A,B). Furthermore, *SLR1*, a critical gene in GA signal transduction, was significantly upregulated (Figure 4C), indicating that the reduced plant height observed in *ddt1* mutants might be due to decreased GA content. To support these findings, we evaluated the protein levels of SLR1, which reflected the changes observed at the mRNA level (Figure 4D). Then, we cloned DDT1 into a pGBKT7 expression vector and SSLR1 into a pGADT7 expression vector, followed by co-transformation into yeast cells. However, yeast two-hybrid assays revealed no direct interaction between SLR1 and DDT1 (Figure 4E), suggesting that DDT1 might indirectly influence *SLR1* expression. 

### 2.5. Role of the DDT1 in Enhancing Drought Stress Tolerance in Rice

In plants, cytochrome P450 enzymes play crucial roles in stress responses [38]. The *DDT1* gene is a member of the cytochrome P450 family. Hence, we examined the expression levels of *DDT1* after subjecting wild-type plants to cold, heat, salt, and drought treatments. The results indicate that salt and drought stress significantly induce the expression of the *DDT1* gene, reaching a peak at 5 h under salt stress and 4 h under drought stress (Figure 5C,D). In contrast, there were no significant changes under cold and heat stress (Figure 5A,B). These experimental findings suggest that the *DDT1* gene may play a role in regulating the response to drought and salt stress in rice.

To further investigate the impact of the *DDT1* gene on drought resistance, we treated the wild-type and *ddt1* mutant rice plants with 20% PEG. The results showed that the leaves of the wild-type plants exhibited severe curling, whereas the mutant leaves displayed no significant changes. After a 7-day recovery period, the survival rate of the mutants was found to be over 60%, compared to less than 20% for the wild type (Figure 6A). This suggests that the mutation in the *DDT1* gene enhances drought tolerance in rice. To further confirm the drought resistance of the mutants, both wild-type and *ddt1* plants were subjected to mannitol treatment in tissue culture, which demonstrated more pronounced growth inhibition in the wild type (Figure 6B). Additionally, we measured the water loss rate of both genotypes, finding that the wild type exhibited a significantly higher rate of water loss compared to the mutant (Figure 6C). We also examined and quantified the behavior of stomata in leaves of both genotypes before and after drought treatment. Microscopic observations indicated that post-drought, the number of fully open and partially open stomata was significantly lower in the mutant, while the number of fully closed stomata was significantly higher compared to the wild type (Figure 6D,E). Furthermore, we analyzed leaf length, leaf width, and stomatal density in both the WT and *ddt1* plants. Our findings indicate that leaf length and width were significantly reduced in *ddt1* compared to the wild type, while stomatal density showed no significant changes (Figure 6F–H). This suggests that the reduced water loss in the mutant under drought conditions may be associated with rapid stomatal closure induced by the drought.

### 2.6. Enhancing Drought Tolerance through Antioxidant Capacity Modulation in ddt1 Mutant 

To investigate whether mutations in the *DDT1* gene enhance drought tolerance by affecting the plant’s antioxidant capacity, this study measured the levels of superoxide anions, hydrogen peroxide, and malondialdehyde (MDA) in wild-type and *ddt1* mutant plants before and after drought treatment. The results showed that after drought stress, the levels of superoxide anions, hydrogen peroxide, and MDA were significantly higher in the wild type than in the mutant (Figure 7A–C). Additionally, this study examined changes in proline content in both genotypes before and after drought treatment, revealing a significant increase in proline levels in the mutant compared to the wild type (Figure 7D). Furthermore, the activities of antioxidant enzymes such as ascorbate peroxidase (APX), peroxidase (POD), superoxide dismutase (SOD), and catalase (CAT) were assessed, with results indicating significantly higher activities in the ddt1 mutant after drought stress (Figure 7E–H). To further validate whether the enhanced drought tolerance of the *ddt1* mutant is mediated through its antioxidant capacity, the expression levels of antioxidant-related genes, including *APX1*, *POD1*, *FeSOD*, and *CATB*, were measured before and after drought treatment. The expression levels of these genes were significantly higher in the mutant compared to the wild type after drought treatment (Figure 7I–L). Collectively, these findings suggest that the mutation in the *DDT1* gene enhances drought tolerance by impacting the plant’s antioxidant capacity.

## 3. Discussion

The significance of rice as a staple food for over half of the global population underscores the critical need to understand and enhance its agronomic traits, such as plant height, which significantly influences yield and lodging resistance. Recent research has illuminated the complex genetic and molecular networks that regulate plant height, focusing on identifying QTLs and genes responsible for this trait. Among these, GA and BR signaling pathways have emerged as key regulatory mechanisms [39]. Moreover, rice’s response to abiotic stresses, such as drought, involves a sophisticated network of molecular mechanisms, where the hormone ABA plays a central role in regulating plants’ water use efficiency and osmotic balance. The synthesis and accumulation of osmoprotectants and the activation of antioxidant defense systems further help the plant mitigate the impacts of stress [24,40]. In this context, the characterization of the rice *dwarf and drought tolerance 1* (*ddt1*) mutant, which corresponds to the previously reported P450 gene CYP96B4/SD37/DSS1 [29,36,37], provides insights into both the regulation of plant height and stress responses. Previous studies have shown that cytochrome P450 enzymes play critical roles in plant growth and stress responses. For instance, cytochrome P450 enzymes are involved in the biosynthesis of various phytohormones, including gibberellins, which are essential for regulating plant height [41,42]. Mutations in cytochrome P450 genes have been linked to altered plant architecture and improved stress tolerance in several plant species. For example, the cytochrome P450 gene CYP78A5 has been implicated in controlling organ size and stress responses in Arabidopsis [43]. Similarly, the CYP714B1 and CYP714B2 genes in rice are involved in gibberellin deactivation, affecting plant height and stress tolerance [44]. The *ddt1* mutant exhibits a distinct dwarf phenotype along with enhanced drought tolerance, which are attributed to changes in GA metabolism and stress response pathways.

The identification of the *DDT1* gene within a critical region influencing plant height and stress responses illustrates the multifunctional nature of some genetic elements in rice. This gene, through its involvement in the biosynthesis and signaling pathways of phytohormones, particularly GA and ABA, illustrates a mechanistic overlap between growth regulation and stress response [45]. The response of *ddt1* mutants to drought stress, particularly through the modulation of antioxidant mechanisms, highlights an adaptive strategy employed by rice. The increased expression of antioxidant enzymes and genes in the mutant suggests an enhanced capacity to mitigate oxidative stress, which is commonly exacerbated under drought conditions [46]. Cytochrome P450 enzymes have been shown to participate in the detoxification of reactive oxygen species (ROS) and the synthesis of secondary metabolites involved in stress defense [47]. The findings that mutants maintain lower levels of reactive oxygen species (ROS) and higher levels of protective osmoprotectants such as proline during stress conditions reflect an intrinsic modification of the cellular defense mechanisms, likely conferred by the mutation in the *DDT1* gene.

Understanding the genetic basis of traits like plant height and drought tolerance provides critical insights for rice breeding programs. The reduced plant height and altered water use efficiency (WUE) are primary factors contributing to the improved drought tolerance in the *ddt1* mutant. This validation supports the potential for using targeted gene editing technologies, such as CRISPR/Cas9, to manipulate *DDT1* function. CRISPR/Cas9 has been successfully employed to edit genes involved in abiotic stress tolerance and growth regulation in various crops [48]. By precisely editing the *DDT1* gene, it may be possible to mitigate the dwarf phenotype while retaining or even enhancing drought tolerance, which could be instrumental in breeding programs aimed at developing high-yielding, stress-resilient rice varieties. The *ddt1* mutant not only offers a potential candidate gene for manipulating plant height but also suggests a link between dwarfism and stress tolerance that could be exploited to develop rice varieties suited for environments prone to drought stress. This dual benefit could significantly impact rice productivity and sustainability, particularly in regions facing increasing climatic variability.

Further research should aim to elucidate the detailed molecular mechanisms by which the *DDT1* gene influences both growth and stress response pathways. Such studies could involve comprehensive analyses of downstream targets of *DDT1*, interactions with other hormonal pathways, and its role in cellular signaling networks. Additionally, field trials and phenotypic analyses under varied environmental conditions would be essential to confirm the laboratory findings and assess their practical implications for crop improvement [49]. 

## 4. Materials and Methods

### 4.1. Plant Material and Growth Conditions

This study utilized the *indica* rice cultivar Shuhui527 as the wild type (WT) and its corresponding mutant line, *ddt1*, which was generated through ethyl methane sulfonate (EMS) mutagenesis. Both the WT and mutant seeds were germinated and grown under natural field conditions in Fuyang (Zhejiang province, coordinates: 119°95′ E, 30°05′ N) and Lingshui (Hainan province, coordinates: 110°02′ E, 18°48′ N), exploiting the optimal growth periods across two distinct climatic zones during the rice growing season.

### 4.2. Genetic Mapping and Mutant Analysis

To identify the genetic basis of the *ddt1* phenotype, we first crossed *ddt1* mutants with the wild type, Shuhui527. F_1_ progenies were backcrossed to the parent *ddt1* to generate an F_2_ population, which was then used for segregation analysis and mapping. A total of 210 F_2_ plants were phenotypically screened, and genotyping was performed using simple sequence repeats (SSRs) and sequence tagged site (STS) markers covering all 12 rice chromosomes. The initial mapping localized the mutation to a region on chromosome 3, which was further refined through additional crosses and finer mapping strategies. The specific primer sequences employed in this study are detailed in Appendix A.

### 4.3. Characterization of DDT1

The candidate region identified was sequenced to pinpoint mutations affecting gene function. *LOC_Os03g04680* was identified as the gene disrupted in the *ddt1* mutant. This gene encodes a cytochrome P450 enzyme. To confirm the function of *LOC_Os03g04680*, we cloned a wild-type copy of the gene into the binary vector pCAMBIA1300 [50] and transformed it into *ddt1* mutants using Agrobacterium-mediated transformation. Transgenic plants were assessed for phenotypic restoration to wild type.

### 4.4. Phenotypic Assessments

Detailed phenotypic analyses were conducted on WT, *ddt1* mutant, and transgenic lines. Measurements included plant height, internode length, grain size, and grain weight. Seeds were soaked at room temperature for 48 h until germination, followed by sowing them in a 96-well plate without bottoms. Seedlings were cultured under a 13 h light/11 h dark photoperiod. For PEG treatment, 14-day-old seedlings were transferred into a culture solution containing 18% (*w*/*v*) PEG6000. After 7–14 days of treatment, recovery was initiated, and physiological responses such as survival rate, water loss rate, and stomatal conductance were recorded.

### 4.5. qRT-PCR Analyses

Quantitative real-time PCR (qRT-PCR) was used to assess the expression levels of *DDT1* and other stress-related genes in various tissues under normal and stress conditions. Total RNA was isolated from various organs using an RNA extraction kit (Cowin Biotech, Taizhou, China). gDNase-treated RNA was reverse transcribed to generate first-strand cDNA using the one step RT Kit (Cowin Biotech). qRT-PCR was performed using SYBR Premix Ex Taq II (TaKaRa Bio, Shiga, Japan) with the CFX96 Real-Time PCR Detection System (Bio-Rad, Hercules, CA, USA). The qPCR conditions were as follows: 95 °C for 3 min and 40 cycles of 94 °C for 30 s, 55 °C for 30 s, and 72 °C for 40 s. The rice *Actin* gene (*LOC_Os03g50885*) was used as an internal control.

### 4.6. Subcellular Localization of DDT1

For subcellular localization studies, the DDT1 coding sequence was fused to the green fluorescent protein (GFP) under the control of the cauliflower mosaic virus 35S promoter (CaMV 35S) in the vector p35S::DDT1-GFP. The construct was transiently expressed in rice protoplasts isolated from young leaves of wild-type plants. Protoplasts were prepared by enzymatic digestion of leaf tissue followed by purification through a sucrose gradient. Transfection of the protoplasts was performed using polyethylene glycol (PEG)-mediated transformation. Twenty hours post-transfection, the protoplasts were observed under a confocal laser scanning microscope to examine the fluorescence emitted by the GFP fusion protein. The specific subcellular localization of the DDT1-GFP fusion protein was determined based on co-localization with organelle-specific markers for the endoplasmic reticulum.

### 4.7. Yeast Two-Hybrid Assay

Clone the full-length DDT1 into the pGBKDT7 vector. Clone the full-length SLR1 into the pGADT7 vector. Co-transform the yeast strain AH109 with DDT1-pGBKT7 and SLR1-pGADT7. Grow yeast cells on -Leu/-Trp or -Leu/-Trp/-His/-Ade media for 4 days.

### 4.8. SDS–PAGE and Western Blot Analysis

Protein samples were fractionated by sodium dodecyl sulfate (SDS)-polyacrylamide gel electrophoresis (PAGE). For Western blot analysis, the proteins were blotted onto a polyvinylidene fluoride membrane and probed with anti-SLR1 (1:500) and anti-ACTIN (1:5000, Cowin Biotech).

## 5. Conclusions

The intricate relationship between plant height, yield, and stress tolerance in rice (as mediated by the *DDT1* gene) offers promising avenues for enhancing crop resilience through genetic interventions. By leveraging advanced genetic and molecular tools, researchers can potentially unlock new dimensions of crop improvement that align with the global food security goals.

## Figures and Tables

**Figure 1 ijms-25-07629-f001:**
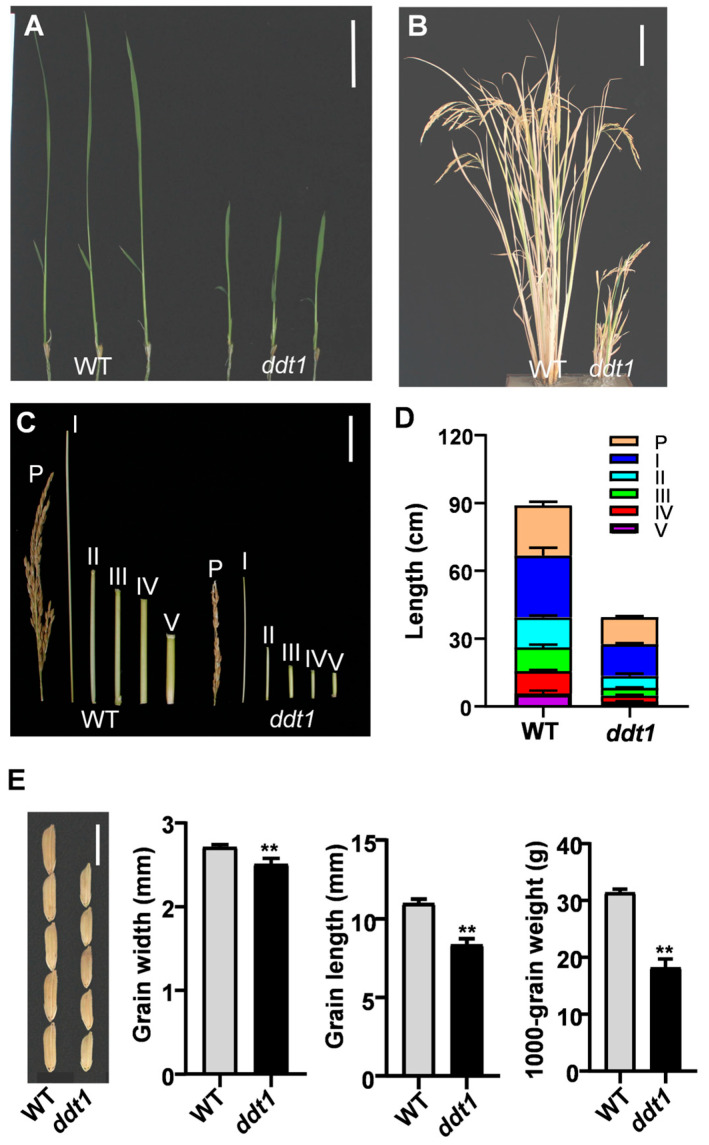
Phenotypic analysis of *ddt1*. (**A**) Plant phenotype of WT and *ddt1* mutant during the seedling stage. Scale bar = 5 cm. (**B**) Whole plant phenotype of WT and *ddt1* mutant during the heading stage. Scale bar = 10 cm. (**C**) Phenotypic comparison of panicles and internodes in WT and *ddt1*. P, I, II, III, IV, and V indicate the panicle, the first, the second, the third, the fourth, and the fifth internode, respectively. Scale bar = 5 cm. (**D**) Statistical analysis of the lengths of panicles and internodes in WT and *ddt1*. (**E**) Grain phenotype, scale bar = 1 cm. ** Indicates statistical significance determined by student *t*-test.

**Figure 2 ijms-25-07629-f002:**
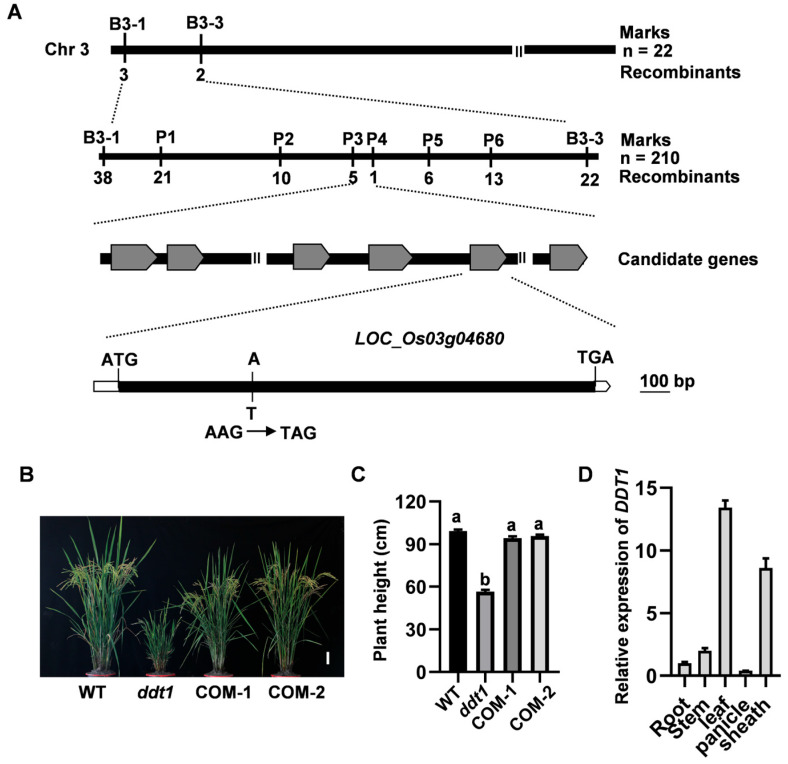
Map-based cloning of *OsDDT1*. (**A**) Map-based cloning of *OsDDT1*. (**B**) Plant phenotype of WT, *ddt1*, and complementary plants during the heading stage. Scale bar = 10 cm. (**C**) Plant height of WT, *ddt1*, and complementary plants during the heading stage. Different letters indicate significant differences. (**D**) Expression patterns of *DDT1*. Data are presented as mean ± SD (*n* = 3). Different letters denote significant differences at *p* < 0.05 (one-way ANOVA, Duncan’s multiple range test). All experiments were repeated at least three times with similar results.

**Figure 3 ijms-25-07629-f003:**
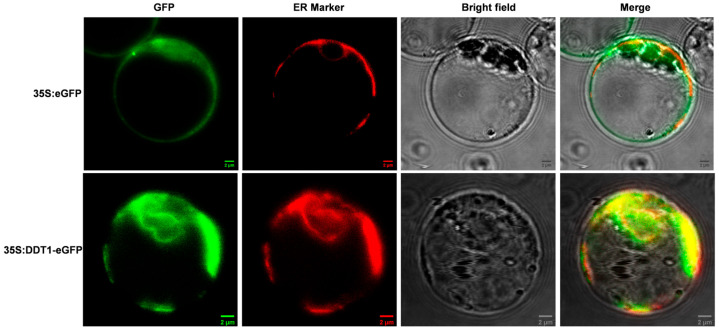
Subcellular localization of DDT1. Scale bar = 2 μm.

**Figure 4 ijms-25-07629-f004:**
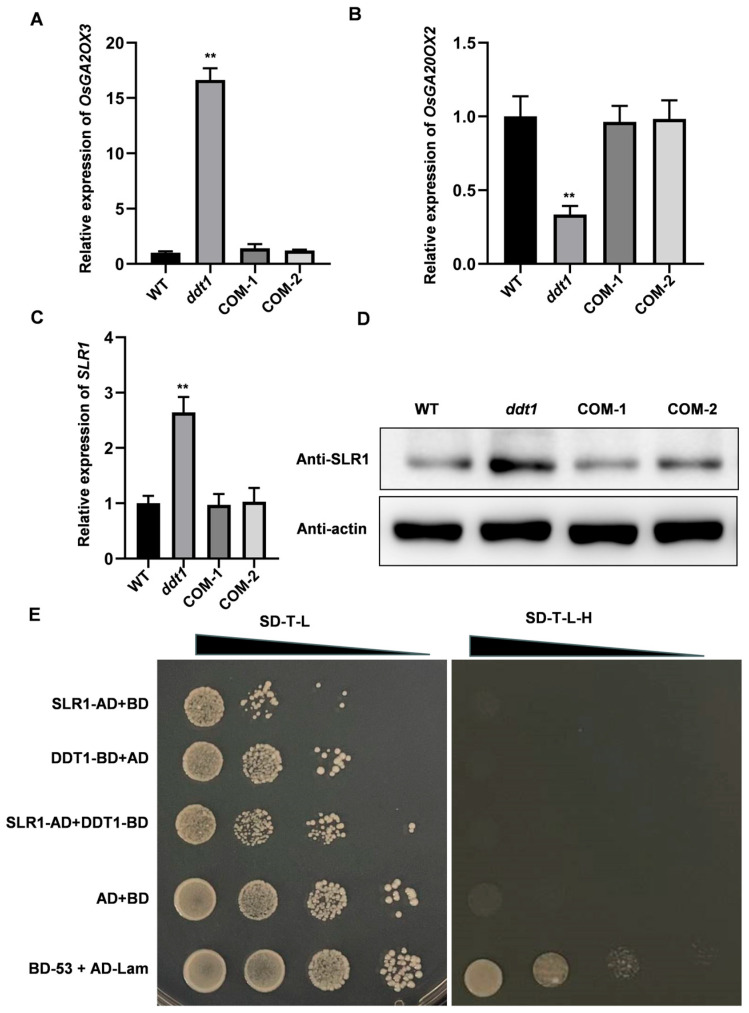
Role of *DDT1* in plant height. (**A**–**C**) Relative expression of *OsGA20X3*, *OsGA20X3*, and *SLR1*. (**D**) SLR1 protein amount in WT, *ddt1* mutant, and complementary plants. (**E**) Y2H assay between SLR1 and DDT1. Data are presented as mean ± SD (*n* = 3). ** Indicates statistical significance determined by student *t*-test.

**Figure 5 ijms-25-07629-f005:**
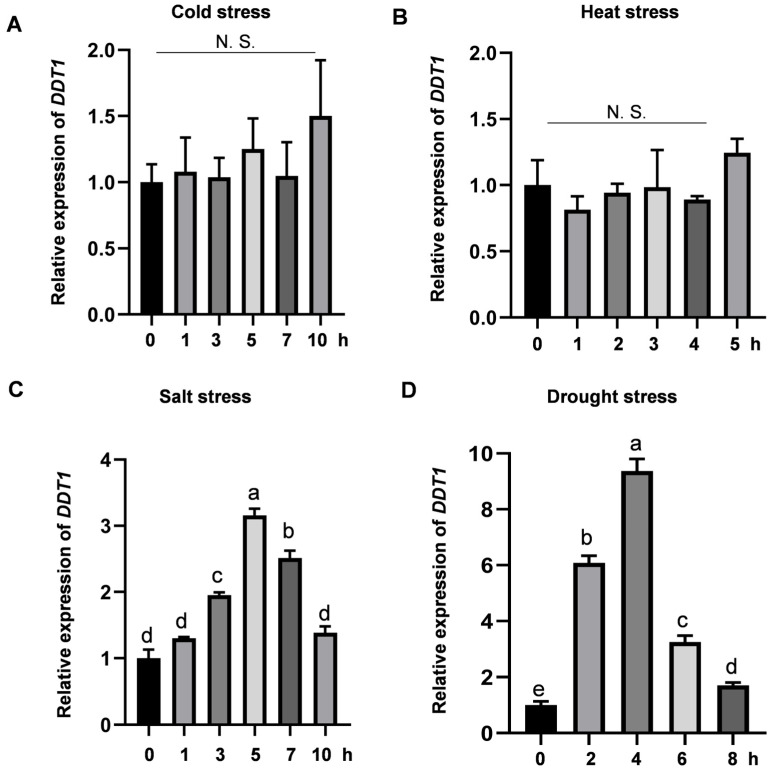
Relative expression of *DDT1* in stress responses. (**A**–**D**) RT-qPCR analysis of *OsDTT1* in rice seedlings treated with 4 °C, 45 °C, 180 mM NaCl and 20%PEG. Data are presented as mean ± SD (*n* = 3). Different letters denote significant differences at *p* < 0.05 (one-way ANOVA, Duncan’s multiple range test). N. S. indicates no significant difference.

**Figure 6 ijms-25-07629-f006:**
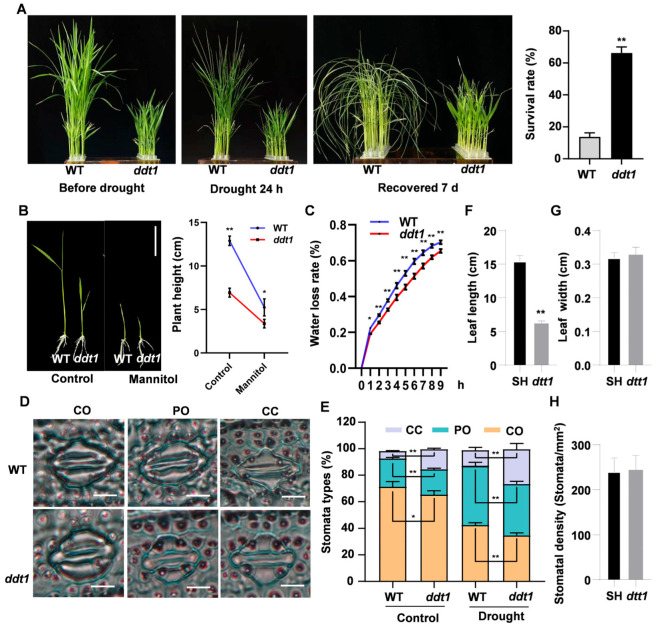
Enhancing drought stress tolerance in *ddt1* mutant. (**A**) Phenotype and survival rate of WT and *ddt1* under drought treatment. (**B**) Phenotype and statistical analysis of plant height between WT and *ddt1* under mannitol treatment. (**C**) Water loss rate of WT and *ddt1*. (**D**) Stomatal observation of WT and *ddt1*. (**E**) Statistical of stomatal opening in WT and *ddt1*. CO, completely open; PO, partially open; CC, completely closed. Leaf length (**F**), leaf width (**G**), and stomatal density (**H**) in WT and *ddt1* plants. Data are presented as means ± SD (*n* = 6), with statistical significance evaluated via Student’s *t*-test (* *p* < 0.05, ** *p* < 0.01).

**Figure 7 ijms-25-07629-f007:**
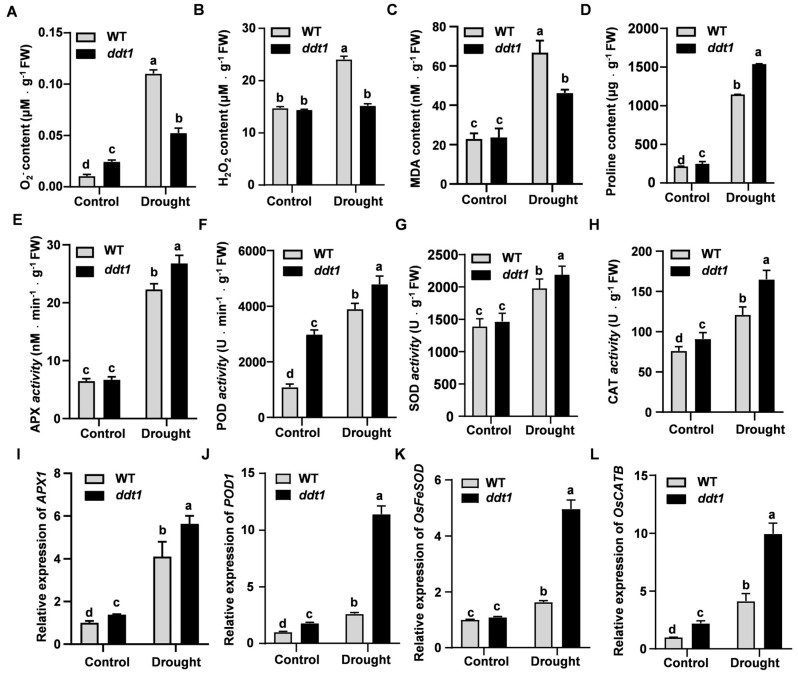
Analysis of antioxidant capacity of *ddt1*. (**A**–**D**) ROS and proline content. (**E**–**H**) Antioxidant enzyme activity. (**I**–**L**) The expression of antioxidase-related genes. Data are presented as mean ± SD (*n* = 3). Different letters denote significant differences at *p* < 0.05 (one-way ANOVA, Duncan’s multiple range test).

## Data Availability

Data are available upon request.

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
