# Peer review of "Characterization of the ddt1 Mutant in Rice and Its Impact on Plant Height Reduction and Water Use Efficiency"

_ijms, 2024, doi:10.3390/ijms25147629_

Round 1

Reviewer 1 Report

Comments and Suggestions for Authors

The authors have identified the dwarf and drought tolerance 1 (ddt1) mutant in rice, which is associated with a cytochrome P450 gene. The ddt1 locus is linked to reduced plant height due to shorter internodes, contributing to increased resistance to lodging. The researchers conducted a series of experiments, including phenotypic analysis of ddt1, map-based cloning of OsDDT1, subcellular localization of DDT1, and an examination of the role of DDT1 in plant height by studying the relative expression of OsGA20X3, OsGA20X3, and SLR1, as well as the Y2H interaction between SLR1 and DDT1. They also analyzed the relative expression of DDT1 in drought stress response. The authors have determined that the ddt1 mutant exhibits improved drought tolerance, characterized by enhanced water retention capabilities and altered physiological responses under water-deficit conditions. After a thorough examination, the reviewer would like to highlight specific concerns that require attention to improve the quality and clarity of the content.

Although the author has conducted a detailed analysis of the ddt1 mutant in rice, the drought tolerance mechanism has been overstated. The ddt1 mutant significantly reduced plant height and overall transpiration area, thereby exhibiting improved water use efficiency (WUE) compared to the wild-type plant. Therefore, it is recommended that the title of the paper be modified to a more general form, such as "Characterization of the ddt1 mutant," and the claims regarding drought tolerance should be toned down. The mutant displays a dwarfed phenotype, which is not directly relevant to agronomic importance, and this gene may not be suitable for deployment in crop improvement programs.

The abstract section requires further expansion and refinement in order to effectively and concisely communicate the key findings, their significance, and their contextualization within the relevant published literature.

The introduction section should be revised to provide more comprehensive information, enhance the relevance of the study, and clearly articulate its significance.

In the result section, the authors have already demonstrated the agronomic performance of the ddt1 mutant. However, it is unclear why they chose to omit important agronomic information, such as grain yield from the report. The phenotypic description of the plant suggests that there may have been a drastic shift in grain yield, which is a critical agronomic trait. Providing a more comprehensive analysis of the mutant's agronomic performance, including the omitted data, would have given readers a more complete understanding of the implications of this genetic modification. The green house yield-related data is important and should be included in the manuscript to provide a comprehensive analysis of the experimental results.

The drought experiment presented does not adequately address the biological context. The mutant plant exhibited a dramatic reduction in biomass and total leaf area, hindering transpiration, and rendering it not directly comparable to the wild-type (WT) plant, which has a larger leaf area and higher stomatal density. This difference likely explains the faster wilting observed in the WT plant compared to the mutant. To fully characterize the stomatal phenotype, the authors should include a graph quantifying the stomatal number in both the WT and mutant lines.

In the figure 6; the significance level is absent in the b, c and e statical graphs

Figure 7 employed a, b, c, and d as indicators of the level of significance in the graphs. However, the manuscript lacks the text that details the significance represented by these indicators.

The discussion section requires strengthening through the incorporation of relevant, previously published literature pertaining to the current study.

The Methodology section requires more detailed explanation and elaboration to fully convey the research approach and procedures undertaken in the study.

The material and methods section lacks detailed descriptions of the plant growth conditions and the experimental drought treatment, including the use of polyethylene glycol (PEG).

The limited duration of a 24-hour drought treatment may not hold significant relevance from an agricultural field perspective. Further justification is needed to establish its applicability and importance in the context of agricultural practices.

Comments on the Quality of English Language

Minor editing of English language required

Author Response

The authors have identified the dwarf and drought tolerance 1 (ddt1) mutant in rice, which is associated with a cytochrome P450 gene. The ddt1 locus is linked to reduced plant height due to shorter internodes, contributing to increased resistance to lodging. The researchers conducted a series of experiments, including phenotypic analysis of ddt1, map-based cloning of OsDDT1, subcellular localization of DDT1, and an examination of the role of DDT1 in plant height by studying the relative expression of OsGA20X3, OsGA20X3, and SLR1, as well as the Y2H interaction between SLR1 and DDT1. They also analyzed the relative expression of DDT1 in drought stress response. The authors have determined that the ddt1 mutant exhibits improved drought tolerance, characterized by enhanced water retention capabilities and altered physiological responses under water-deficit conditions. After a thorough examination, the reviewer would like to highlight specific concerns that require attention to improve the quality and clarity of the content.

  1. Although the author has conducted a detailed analysis of the ddt1 mutant in rice, the drought tolerance mechanism has been overstated. The ddt1 mutant significantly reduced plant height and overall transpiration area, thereby exhibiting improved water use efficiency (WUE) compared to the wild-type plant. Therefore, it is recommended that the title of the paper be modified to a more general form, such as "Characterization of the ddt1 mutant," and the claims regarding drought tolerance should be toned down. The mutant displays a dwarfed phenotype, which is not directly relevant to agronomic importance, and this gene may not be suitable for deployment in crop improvement programs.

R: We acknowledge that the enhanced water use efficiency (WUE) due to the reduced plant height and transpiration area in the ddt1 mutant contributes significantly to its observed drought tolerance. To address this, we revise the manuscript to emphasize that the reduced plant height and altered WUE are primary factors contributing to the improved drought tolerance in the ddt1 mutant.

We agree that the current title may overstate the drought tolerance mechanism. We had changed the title to " Characterization of the ddt1 Mutant in Rice and Its Impact on Plant Height Reduction and Water Use Efficiency".

We understand your concern regarding the agronomic importance of the dwarf phenotype. However, we believe that the gene associated with DDT1 holds potential for improving drought tolerance through gene editing techniques. By leveraging this gene, it is possible to develop germplasm with better agronomic traits and enhanced drought resistance.

  1. The abstract section requires further expansion and refinement in order to effectively and concisely communicate the key findings, their significance, and their contextualization within the relevant published literature.

R: We acknowledge the feedback regarding the abstract. We have expanded the abstract as per your request, as follows: “Rice (Oryza sativa L.), a fundamental global staple, nourishes over half of the world’s population. The identification of the ddt1 mutant in rice through EMS mutagenesis of the indica cultivar Shuhui527 revealed a dwarf phenotype, characterized by reduced plant height, smaller grain size, and decreased grain weight. Detailed phenotypic analysis and map-based cloning pinpointed the mutation to a single-base transversion in the LOC_Os03g04680 gene, encoding a cytochrome P450 enzyme, which results in a premature termination of the protein. Functional complementation tests confirmed LOC_Os03g04680 as the DDT1 gene responsible for the observed phenotype. We further demonstrated that the ddt1 mutation leads to significant alterations in gibberellic acid (GA) metabolism and signal transduction, evidenced by the differential expression of key GA-related genes such as OsGA20OX2, OsGA20OX3, and SLR1. The mutant also displayed enhanced drought tolerance, as indicated by higher survival rates, reduced water loss, and rapid stomatal closure under drought conditions. This increased drought resistance was linked to the mutant's improved antioxidant capacity, with elevated activities of antioxidant enzymes and higher expression levels of related genes. Our findings suggest that DDT1 plays a crucial role in regulating both plant height and drought stress responses. The potential for using gene editing of DDT1 to mitigate the dwarf phenotype while retaining improved drought resistance offers promising avenues for rice improvement.”

  1. 3. The introduction section should be revised to provide more comprehensive information, enhance the relevance of the study, and clearly articulate its significance.

R: Thank you for your valuable feedback. We revised the introduction to include more comprehensive information, enhance the relevance of the study, and clearly articulate its significance.

  1. In the result section, the authors have already demonstrated the agronomic performance of the ddt1 mutant. However, it is unclear why they chose to omit important agronomic information, such as grain yield from the report. The phenotypic description of the plant suggests that there may have been a drastic shift in grain yield, which is a critical agronomic trait. Providing a more comprehensive analysis of the mutant's agronomic performance, including the omitted data, would have given readers a more complete understanding of the implications of this genetic modification. The green house yield-related data is important and should be included in the manuscript to provide a comprehensive analysis of the experimental results.

R: Thank you for your insightful comments regarding the agronomic performance of the ddt1 mutant. We acknowledge the importance of agronomic performance in evaluating the mutant's utility in agricultural contexts. Consequently, we have included additional data on grain yield per plant in the manuscript (Fig. S1).

  1. The drought experiment presented does not adequately address the biological context. The mutant plant exhibited a dramatic reduction in biomass and total leaf area, hindering transpiration, and rendering it not directly comparable to the wild-type (WT) plant, which has a larger leaf area and higher stomatal density. This difference likely explains the faster wilting observed in the WT plant compared to the mutant. To fully characterize the stomatal phenotype, the authors should include a graph quantifying the stomatal number in both the WT and mutant lines.

R: Thank you for your insightful feedback, which has greatly influenced our revisions. We have added statistical graphs depicting stomatal density, leaf length, and width for both wild-type and mutant plants. Our findings indicate that leaf length and width were significantly reduced in ddt1 compared to the wild type, while stomatal density showed no significant changes (Fig. 6F-H).

  1. In the figure 6; the significance level is absent in the b, c and e statical graphs

R: We have supplemented the significance levels in graphs b, c, and e.

  1. Figure 7 employed a, b, c, and d as indicators of the level of significance in the graphs. However, the manuscript lacks the text that details the significance represented by these indicators.

R: We have provided detailed annotations for the significance level indicators in the statistical charts.

  1. The discussion section requires strengthening through the incorporation of relevant, previously published literature pertaining to the current study.

R: Thank you for your suggestion. We have enhanced our study by incorporating relevant literature.

  1. The Methodology section requires more detailed explanation and elaboration to fully convey the research approach and procedures undertaken in the study.

R: Thank you for your valuable feedback regarding the Methodology section of our manuscript. We had expanded the Methodology section to provide a thorough description of the experimental design.

  1. The material and methods section lacks detailed descriptions of the plant growth conditions and the experimental drought treatment, including the use of polyethylene glycol (PEG).

R: We have supplemented the corresponding experimental methods.

  1. The limited duration of a 24-hour drought treatment may not hold significant relevance from an agricultural field perspective. Further justification is needed to establish its applicability and importance in the context of agricultural practices.

R: We appreciate your feedback on the duration of the drought treatment in our study. We understand that a 24-hour drought treatment may seem limited from an agricultural field perspective. However, our choice of a 24-hour drought treatment was based on specific scientific and methodological considerations, which we believe are crucial for the following reasons:

The 24-hour drought treatment was employed as an initial screening tool to rapidly identify and evaluate the physiological and genetic responses of the rice plants to drought stress under controlled conditions. This duration allows us to observe immediate and acute responses, which are often indicative of the plant's overall stress tolerance mechanisms.

Short-term drought treatments are valuable for studying molecular and physiological changes that occur in the early stages of drought stress. These changes can include alterations in gene expression, hormone signaling pathways, and antioxidant enzyme activities. By focusing on a 24-hour period, we can capture these early responses, which are critical for understanding the underlying mechanisms of drought tolerance.

In agricultural fields, plants can experience transient drought events where water availability fluctuates rapidly, especially in areas with irregular rainfall patterns or during sudden dry spells. Understanding how plants respond to short-term drought stress can provide insights into their ability to withstand these transient events and recover when water becomes available again.

The findings from the 24-hour drought treatment serve as a foundation for designing longer-term and more field-relevant drought experiments. By first identifying key genetic and physiological responses in a controlled environment, we can better tailor subsequent experiments to address field conditions, thereby ensuring a more focused and efficient use of resources.

In summary, while the 24-hour drought treatment may not directly mimic prolonged drought conditions in agricultural fields, it provides essential insights into the immediate stress responses and lays the groundwork for future studies. These insights are crucial for developing strategies to enhance drought tolerance in rice and other crops under both short-term and long-term drought scenarios. We plan to build upon these findings in future research to explore the effects of extended drought periods and validate our results in field trials.

Reviewer 2 Report

Comments and Suggestions for Authors

The present manuscript analyzes the rice ddt1 mutant that exhibits a dwarf phenotype compared with the wild type plants, but with well expressed drought tolerance. The authors obtained that DDT1 gene localized in chromosome 3 of the rice has a main role in rice growth regulation and stress response to drought. An effect on gibberellin acid synthesis and increased antioxidant activity that reflects to cellular defense mechanism have been reported.

The obtained information in the manuscript is useful because the enhance adaptation of a major agricultural crops such as rice to drought and other constantly changing abiotic stress factors is of great importance.

The English of the manuscript is easy to read and the work is understandable.

The results and the discussion are well presented. Some more details need to be added for the methods used in the study. The conclusions are well done.

Some clarifications should be made.

1.     The authors should define more clearly the aim of the study in the part “Introduction”. The text given on lines 70-76 is like conclusion.

2.     Please, mark the statistical difference values under the Figure 1. The same concerns data on Figure 2.

3.     What are the complementary plants marked on the figures as COM-1 and COM-2? Please, mention in the text.

4.     Some more details should be given about the yeast two-hybrid assay (Y2H) mentioned on lines 156-158, as well as for the results presented on Figure 4 (E)

5.     On Figure 5 the values of statistical significances should be marked.

6.  The title of Figure 6 should be corrected and the value of statistical significances on 6 (B) should be added.

7.     The text on lines 219-221 is not necessary.

8.     In the part Material and Methods to the text under the subheading 4.2. Genetic Mapping and Mutant Analysis, the authors should point out from where is the information regarding the repeat types of the (SSRs) and sequence tagged sites (STSs) markers? Did they use any WEB database?

9.     On lines 290 the authors should cite the reference about the binary vector pCAMBIA1300.

10.  Please, insert some details about the qRT-PCR conditions on lines 300-302.

Author Response

The present manuscript analyzes the rice ddt1 mutant that exhibits a dwarf phenotype compared with the wild type plants, but with well expressed drought tolerance. The authors obtained that DDT1 gene localized in chromosome 3 of the rice has a main role in rice growth regulation and stress response to drought. An effect on gibberellin acid synthesis and increased antioxidant activity that reflects to cellular defense mechanism have been reported.

The obtained information in the manuscript is useful because the enhance adaptation of a major agricultural crops such as rice to drought and other constantly changing abiotic stress factors is of great importance.

The English of the manuscript is easy to read and the work is understandable.

The results and the discussion are well presented. Some more details need to be added for the methods used in the study. The conclusions are well done.

Some clarifications should be made.

1.The authors should define more clearly the aim of the study in the part “Introduction”. The text given on lines 70-76 is like conclusion.

R: Thank you for your suggestion. We have revised the Introduction as requested.

  1. Please, mark the statistical difference values under the Figure 1. The same concerns data on Figure 2.

R: Thank you for your suggestion. We have supplemented the statistical significance values below Figures 1 and 2.

3.What are the complementary plants marked on the figures as COM-1 and COM-2? Please, mention in the text.

R: Thank you for your suggestion. COM-1 and COM-2 have been added to the main text.

  1. Some more details should be given about the yeast two-hybrid assay (Y2H) mentioned on lines 156-158, as well as for the results presented on Figure 4 (E)

R: We have supplemented more details about Y2H in both the Results and Methods sections.

5.On Figure 5 the values of statistical significances should be marked.

R: We have supplemented the statistical significance analysis on Figure 5.

6.The title of Figure 6 should be corrected and the value of statistical significances on 6 (B) should be added.

R: We have changed the title of Figure 6 to “Enhancing drought stress tolerance in ddt1 mutant” and the value of statistical significances on Fig. 6B have been added.

7.The text on lines 219-221 is not necessary.

R: We have removed the text from lines 219-221.

8.In the part Material and Methods to the text under the subheading 4.2. Genetic Mapping and Mutant Analysis, the authors should point out from where is the information regarding the repeat types of the (SSRs) and sequence tagged sites (STSs) markers? Did they use any WEB database?

R: Thank you for your query. The SSRs and sequence tagged sites (STSs) markers used in our study were sourced from our own laboratory.

9.On lines 290 the authors should cite the reference about the binary vector pCAMBIA1300.

R: Thank you for pointing this out. We have now included the citation for the binary vector pCAMBIA1300 as suggested.

10.Please, insert some details about the qRT-PCR conditions on lines 300-302.

R: We have supplemented detailed information regarding the qRT-PCR conditions.